# The Influence of Standardized Residency Training on Trainees’ Willingness to Become a Doctor: A Comparison between Traditional Chinese Medicine and Western Medicine

**DOI:** 10.3390/ijerph16173017

**Published:** 2019-08-21

**Authors:** Junwen Yang-Huang, Wenji Qian, Kan Zhang, Lu Shi, Jiayan Huang

**Affiliations:** 1Key Laboratory of Health Technology Assessment, National Health Commission, School of Public Health, Fudan University, Shanghai 200032, China; 2The Dean’s Office, Fudan University Shanghai Cancer Center, Shanghai 200032, China; 3Science and Education Department, Shanghai Municipal Health Commission, Shanghai 200125, China; 4Department of Public Health Sciences, Clemson University, Clemson, SC 29634, USA

**Keywords:** Traditional Chinese Medicine, standardized residency training, satisfaction survey, willingness change

## Abstract

A standardized residency training program (SRT) was launched in Shanghai in 2010, for both Western Medicine (WM) and Traditional Chinese Medicine (TCM). This study aimed to assess whether the program impacted trainees’ willingness to become a doctor and whether the program had different effects among WM and TCM trainees. A structured questionnaire was distributed to 2114 trainees to assess their perceptions and satisfaction with the program and their willingness to become a doctor after the exposure to the program. The trainees’ characteristics were compared between WM and TCM specialties using X^2^ tests. The potential factors associated with trainees’ perception of the program and willingness to become a doctor after the exposure to the SRT program were assessed by multiple linear and logistic regression models. Compared to WM trainees (*n* = 1853), TCM trainees (*n* = 261) would be more likely to become doctors if there were no SRT program (*p* = 0.003). Both individual and program-wide (different specialties) factors contributed to trainees’ perception, overall satisfaction, and willingness. Only specialty played an independent role in the associations with all three outcome variables. Inequality of characteristics between TCM and WM trainees reduced TCM trainees’ willingness to become a doctor after the exposure to the SRT program.

## 1. Introduction

Despite the significant growth of Western Medicine (WM), Traditional Chinese Medicine (TCM) still plays an important role in Chinese health care [1]. In China, TCM services are provided in all levels of health institutions in both urban and rural areas [2]. In 2014, TCM accounted for 15.6% of China’s total outpatient visits, including both in TCM-specialized health institutions and WM health institutions [3]. The 13th Five-Year Plan for Economic and Social Development of China further emphasizes the development of TCM from 2016 to 2020. The goal is to provide integrated TCM services in more than 85% of urban community health service centers and more than 70% of rural township hospitals [4].

However, it is uncertain whether China has sufficient TCM doctors to fulfill these ambitious goals. Over the past decade, a trend of declining interest among medical graduates in pursuing careers in clinical practice has persisted [5]. From 2005 to 2014, the proportion of physicians aged 25–34 years decreased from 31.3% to 22.6%, and the proportion of physicians aged 60 years and older increased from 2.5% to 11.6% [5]. Certain specialties, such as pediatrics, suffer more from the shortage of medical doctors than others. In the future, a rapidly aging society will only aggravate the problem.

China is reforming its medical education to alleviate the doctor shortage. One focus of current reforms is the residency training program. Medical education in China consists of medical school, graduate medical education, and continuing education [6]. The residency training program, regarded as an important part of graduate medical education, was first introduced in China in 1988. Shanghai was one of the pilot cities to implement the residency training program [7]. Since then, Shanghai has continued to be a pioneer city to promote residency training. In February 2010, Shanghai implemented the Standardized Residency Training (SRT) program, for both WM and TCM graduates, which aims to establish unified training standards at the municipal level. Since then, all the public hospitals and medical institutions in Shanghai would only hire doctors with qualifications from the SRT program, and all medical school graduates need to enroll in the program before practicing medicine in Shanghai [8,9].

This new SRT program has many innovative initiatives compared to the previous residency training program (non-SRT program) (Table 1). First, the program changed the identity of residents. They are now regarded as interns rather than the formal staff of the hospital, and they need to look for a new job after graduation from the SRT program. Second, the SRT program in Shanghai is the first standardized training program in China for TCM graduates to practice traditional examination methods as well as multiple clinical skills. Third, the program established differentiated lengths of training for residents with different academic qualifications. Bachelor’s, master’s, and doctorate graduates were assigned to take the SRT program for three years, two years, and one year, respectively. Because the residents with a master’s or a doctorate degree already have clinical practice experience during medical school study. In addition, when enrolling in the SRT program, residents can apply for different sub-specialties under WM or TCM as their training direction. Each sub-specialty has its training program according to the relevant regulations formulated by the National Health Commission of the People’s Republic of China and the National Administration of Traditional Chinese Medicine of the People’s Republic of China. Although the rotating departments may be different for each sub-specialty, the overall setting of rotating time, objective and graduation examination are the same. Residents can also choose the training hospital they would like to enroll, but they need to pass the hospital-level entrance examinations. Lastly, a joint meeting system is developed to manage the SRT program at the municipal level.

Preliminary evaluations for Shanghai’s SRT program focused on assessing resident trainees’ clinical skills [10,11,12]. Results have shown that the SRT program has achieved the expected goals of encouraging trainees’ learning initiatives and improving their clinical skills. However, some studies have shown concerns about trainees’ reduced willingness to become a doctor after the launch of the SRT program [13]. Trainees were concerned about long training time, low pay during the training, and increased competition for securing a job after the training [14]. However, the existing studies did not consider the potential differences between WM and TCM trainees. Assessments of TCM trainees’ attitudes toward the SRT program and their willingness to become a doctor after the program will be valuable for the education of TCM doctors and the development of TCM in China.

This study hypothesizes that the implementation of the SRT program may have a negative impact on residents’ willingness to practice medicine. This research aims to find out whether the SRT program has an impact on residents’ willingness to practice medicine, especially focusing on the difference between WM and TCM residents. Additionally, we hope to find out the factors related to change of the willingness and accordingly make suggestions to promote the SRT program and help reduce the influence.

## 2. Materials and Methods

### 2.1. Ethics Statement

The study has been approved by the Medical Ethical Committee at Fudan University (registration number: IRB00002408 & FWA00002399). The number for the study is IRB#2013-10-0468. Written informed consent was obtained from all participants.

### 2.2. Study Design

The study is a cross-sectional study to evaluate the trainees’ attitudes toward the SRT program. Invitations for the study were made to all trainees who enrolled in the Shanghai SRT program in 2013. Data on trainees’ attitudes, their satisfaction toward the SRT program, and the change in their willingness to become a doctor after the exposure to the SRT program were collected in June 2014.

### 2.3. Study Sample

By 2013, 44 hospitals had met the requirements for training bases and held the qualifications to carry out the SRT program, including 32 tertiary institutions, 7 secondary institutions, and 5 TCM-specialized institutions. This study included all 2283 trainees in these 44 hospitals under 27 specialties in the Shanghai SRT program. A structured questionnaire was used to investigate trainees’ attitudes toward the SRT program and their willingness to become a doctor after the exposure to the SRT program. In 2014, just before the end of the first study-year, the questionnaires were distributed to the residents online. Residents signed the online informed consent before filling in the questionnaire. The data were collected anonymously. Of all included trainees, 2114 completed the questionnaire survey. The response rate was 92.60%.

### 2.4. Questionnaire

Questions included domains of perception of the SRT program, overall satisfaction toward the SRT program, and willingness to become a doctor after or without the current SRT program. Perception of the SRT program was assessed by the question, “How do you consider the necessity of Shanghai SRT program?” in a five-point Likert scale ranging from 1 (very unnecessary) to 5 (very necessary). Overall satisfaction toward the SRT program was also assessed using a five-point Likert scale ranging from 1 (very unsatisfied) to 5 (very satisfied). Two items in the questionnaire were used to assess the trainees’ willingness to become a clinical doctor: “How strong is your willingness to become a clinical doctor under the new SRT program?” (very strong, strong, undecided, weak) and “How strong would be your willingness to become a clinical doctor if there were no SRT program?” (very strong, strong, undecided, weak). Scores were assigned to the possible answers for the two questions regarding trainees’ willingness to become a clinical doctor (very strong = 4, strong = 3, undecided = 2, weak = 1) [15,16]. The willingness change score was calculated by the formula: willingness change score = willingness score without the SRT program-willingness score after the SRT program. Later, an outcome variable dichotomized as willingness decrease/stable (willingness change score ranges from −3 to 0) (reference group) and willingness increase (willingness change score ranges from 1 to 3) was used for the analysis. Increased willingness if there where no SRT program was set as 1, and vice versa 0.

### 2.5. Data Analyses

First, frequency tables and cross-tabulations were used to explore the characteristics of the study population. Trainees were compared in groups of TCM specialties and WM specialties. 8 TCM sub-specialties (e.g., internal medicine of TCM, gynecology of TCM, osteology and traumatology of TCM, pediatrics of TCM) were categorized into TCM specialties, and 19 WM sub-specialties (e.g., internal medicine, surgery, general practice, pediatrics) were categorized into WM specialties. Chi-square tests were used to test the differences of characteristics between the two groups. Second, we used a multiple linear regression model to explore factors associated with trainees’ perception of and satisfaction toward the SRT program. We used a multiple logistic regression model to explore whether the factors would increase residents’ willingness to practice medicine or not. Trainees’ gender, graduation region, educational level, clinical work experience, physician’s license acquisition, and background (TCM and WM specialties) were considered as the potential factors in the models. The univariate models included each potential factor separately. The multivariate models included all the potential factors. Data preparation and statistical analyses were performed using IBM SPSS Statistics for Windows, Version 21.0. Armonk, NY: IBM Corp. A significance level of *p* < 0.05 was used to indicate significant associations.

## 3. Results

### 3.1. Characteristics of the Study Population

Among 2114 trainees, 261 were trained in TCM specialties (12.3%) and 1853 were trained in WM specialties (87.7%). The mean age of trainees was 26.55 ± 2.36.

Compared to WM-specialty trainees, TCM-specialty trainees were more likely to be female (72.0%), graduate from Shanghai region (57.9%), and have a bachelor’s or master’s degree (95.4%). About 40.6% of TCM-specialty trainees had clinical experience before they enrolled in the SRT program, while in WM-specialty trainees, 36.9% of them had clinical experience. In both WM and TCM specialties, around half of the trainees had already received their physician’s licenses before they enrolled in the SRT program. There were no significant differences in clinical experiences and physician’s licenses between TCM-specialty trainees and WM-specialty trainees (Table 2).

### 3.2. Trainees’ Perception of and Overall Satisfaction toward the SRT Program

Among WM-specialty trainees, 10.5% of them considered the SRT program very necessary for their medical training, while only 6.1% of TCM-specialty trainees considered it very necessary (Table 3). About 5.6% of WM-specialty trainees were very satisfied with the SRT program, while only 2.7% of TCM-specialty trainees scored the program as very satisfactory. The average satisfaction score toward the SRT program in WM-specialty trainees was significantly higher (mean = 3.26, SD = 0.89) than the score in TCM-specialty trainees (mean = 3.03, SD = 0.91) (*p* < 0.001). Around a quarter of TCM-specialty trainees would have increased willingness to practice clinical medicine if there were no SRT program, while only 17% of WM-specialty trainees would have increased willingness if there were no SRT program (*p* = 0.003).

### 3.3. Potential Factors of Trainees’ Perception of, Overall Satisfaction toward the SRT Program, and Change in Willingness to Become a Doctor after the Exposure to SRT Program

In univariate models, gender was not significantly associated with any outcome variables (Table 4). Trainees’ age, educational level, clinical work experience, doctor license, and specialty were associated with all three outcome variables (Table 4). Graduation region was associated with trainees’ perception of the SRT program and willingness to become a doctor after the exposure to SRT program (*p* < 0.05), while its association with trainees’ overall satisfaction toward the SRT program was not significant (*p* = 0.547).

In multivariate models, the clinical working experience was not significantly associated with any outcome variables (Table 4). Trainees’ age was only associated with their perception of the SRT program (*p* = 0.036). Compared with trainees without a physician’s license, those who had already acquired one had a higher chance of increased willingness if there were no SRT program (OR = 1.64, 95% CI: 1.10, 2.46). The graduation region was associated with trainees’ perception of the SRT program (*p* < 0.001) and the change in willingness after the exposure to the SRT program (*p* < 0.001). Trainees’ educational level was associated with the perception of the SRT program (*p* < 0.001 for master’s degree and *p* = 0.003 for doctorate) and overall satisfaction toward the SRT program (*p* = 0.003 for master’s degree and *p* = 0.016 for doctorate). Trainees’ specialty was associated with all three outcomes in the multivariate models. Compared with TCM-specialty trainees, WM-specialty trainees had a better perception of the SRT program (*p* = 0.004) and higher overall satisfaction toward the SRT program (*p* < 0.001). Compared with TCM-specialty trainees, WM-specialty trainees were less likely to show an increase in willingness if there were no SRT program (OR = 0.70, 95%CI: 0.51, 0.97), which means that TCM-specialty trainees would be more likely to become a clinical doctor if there were no SRT program for them.

## 4. Discussion

The present study suggested that after the exposure to SRT program, perception of the SRT program, overall satisfaction toward the SRT program, and change in willingness to become a clinical doctor varied consistently between WM and TCM specialties. We found that WM-specialty trainees tended to view the Shanghai SRT program more favorably than TCM-specialty trainees did. Correspondingly, TCM-specialty trainees would show a more significant increase in willingness if there were no SRT program. Other individual (age, graduation region, educational level, clinical work experience, doctor license) and program-wide (different specialties) factors were not consistently associated with all outcome variables.

We found significant differences in gender, education background, and graduation region between TCM-specialty trainees and WM-specialty trainees, which may result from different career expectations. The situations of gender inequality and lower educational level of TCM trainees were not limited to the Shanghai area, but rather a common phenomenon across China TCM-specialty programs. There are more female TCM-specialty trainees than male trainees [17]. In fact, in the past decade, the proportion of female residents has always been higher than that of male residents by nearly 20% in Shanghai [18].

TCM-specialty trainees consistently rated the Shanghai SRT program more negatively than WM-specialty trainees, after adjusting for all potential factors of outcome variables. This suggested that TCM-specialty trainees were less satisfied with the newest residency training program. This may be because TCM-specialty trainees experienced more significant changes in the training system. Before the SRT program in Shanghai, TCM-specialty graduates were usually locked down at positions in the TCM institutions and trained mostly in TCM examination methods. The current SRT program is the first standardized training program requiring TCM-specialty trainees to practice traditional examination methods as well as multiple clinical skills [19]. The current program emphasized the integration of TCM and WM, under the 13th Five-Year Plan for Economic and Social Development of China. The leap between the former training program and the Shanghai SRT program of TCM is much larger than the alteration in the WM training program. This finding is consistent with previous research which reported lower satisfaction from minority-medicine trainees toward the newest standardized training program [20,21]. In addition, having a lower income than WM residents may also be a reason for lower satisfaction [22]. Additional research is needed to assess the change of TCM-specialty trainees’ satisfaction toward the SRT program.

In the univariate model, all potential factors except gender were associated with trainees’ perception of, overall satisfaction toward the SRT program, and change in willingness to become a doctor after the exposure to SRT program. Senior trainees, trainees who graduated from the Shanghai region, trainees with a higher educational level, and trainees with a physician’s license or work experience had lower satisfaction toward the SRT program and would be more inclined to become a doctor if there were no SRT program. This suggested that trainees with more skills or advantages when they enrolled in the SRT program were less in favor of standardized training. Trainees with more advantages may feel that the SRT program prolonged their path to secure positions in medical institutions. Besides, the graduation region was associated with the perception of the SRT program and an increase in willingness. There was existing evidence showing that trainees dropped out of the SRT program to land a job outside Shanghai [6,23]. The possibility of landing a job outside Shanghai may be even higher if the trainee has already attained a physician’s license. Furthermore, the change of residents’ identity in the hospital may also contribute to a change in their willingness. Before the SRT became compulsory in Shanghai, a medical graduate who successfully found a job became an employee of a hospital. Such a mechanism was still the main path to employment for graduates at the national level, which attracted non-Shanghai graduates to leave Shanghai to find permanent positions immediately [24]. Furthermore, as TCM-specialty trainees usually have a lower educational level than WM-specialty trainees, TCM-specialty trainees may have the lower opportunity cost of landing a job outside Shanghai than WM-specialty trainees. Before SRT programs are implemented nationally, the outflow of TCM talents would be one of the major obstacles for the development of TCM in Shanghai.

Besides, a TCM general practitioner (GP) training was launched in the SRT program [25]. This featured program aims to train TCM GPs serving in community health service centers. The program not only provides extra employment channels for TCM trainees but also cultivates GPs for primary healthcare. This training effectively improves the service capacity in primary healthcare institutions in Shanghai and meets the strategy of the “hierarchical medical system” proposed by the national government.

Data in this paper was collected at the end of the first year after the implementation of the SRT program. The possible reason for the low perception and satisfaction could be the immature institutions and measures, residents’ ignorance of policies and requirements of the SRT program. In the current study, it was found that the perception and satisfaction improved with the promotion of the SRT program [26]. The SRT program in Shanghai set up an expert group and a joint meeting system. The expert group was responsible for periodic inspection on all training bases, which helped to find the problems of the program. The joint meeting system would discuss and solve these problems promptly. This mechanism helped the program to adjust continuously to meet the developing requirements of both trainees and the healthcare system. Therefore, the SRT program became more accepted by trainees in both the TCM specialty and the WM specialty.

This study still has several limitations. First, the career development of the present trainees after the SRT program was not known when the project was launched. There is no existing information to illustrate the trainees’ choices after the SRT program. Future studies are recommended to further explore trainees’ career development. Comparison between trainees’ actual career choices and their change in willingness after the exposure to SRT program would give a more comprehensive understanding of the effect of the SRT program in terms of manpower resources in the TCM-specialty. Second, the present study is the first large-scale study assessing trainees’ perception of and satisfaction toward the Shanghai SRT program, but certain subgroups in the TCM specialty were still relatively small. The project was conducted cross-sectionally on trainees of the class of 2013. Comparisons between trainees from each year or a longitudinal satisfaction evaluation need to be addressed by future research.

## 5. Conclusions

Inequality of gender, educational level, and graduation region existed between TCM-specialty trainees and WM-specialty trainees in the Shanghai SRT program. The inequality contributed to trainees’ perception of and satisfaction with the SRT program, and thus reduced TCM-specialty trainees’ willingness to become a doctor after the exposure to the SRT program. This research highlighted that specialty difference between TCM and WM played an important and independent role in the associations with trainees’ perception of the SRT program, satisfaction toward the SRT program, and their change in willingness to become a doctor after the exposure to SRT program. Policymaking regarding the SRT program needs to take into consideration the risk that certain groups’ willingness to practice medicine could be reduced by this training program. Future policies are needed to enhance the connection between medical education and a systematic residency training program, which will lead to less inequality between trainees with different backgrounds.

## Figures and Tables

**Table 1 ijerph-16-03017-t001:** Difference between SRT program and non-SRT program.

Items	Non-SRT Program	SRT Program
Residents’ identity	An employee in the hospital	An intern in the hospital
Training specialty	WM	WM & TCM
Rotation	Being locked in a specialty	Rotating in different specialties
Length of training	Unified	Differentiated according to academic qualifications
Unified training rules	None	Unified standards at the municipal level
Unified management mechanism	None	Joint meeting system
Graduation examination	Organized by hospitals	Organized at the municipal level
Conditions for practicing medicine	Physician’s practice license	Physician’s practice license & Graduation certificate of the SRT program

SRT = Standardized Residency Training; WM = Western Medicine; TCM = Traditional Chinese Medicine.

**Table 2 ijerph-16-03017-t002:** Characteristics of the trainees (*N* = 2114).

Items	Specialty		Total *N* = 2114	*p*-Value ^b^
	WM-Specialty ^a^	TCM-Specialty ^a^		
	*N* = 1853 (87.7)	*N* = 261 (12.3)		
Gender *N* (%)				0.003
Male	694 (37.5)	73 (28.0)	767 (36.3)	
Female	1159 (62.5)	188 (72.0)	1347 (63.7)	
Graduation Region *N* (%)				<0.001
Shanghai	465 (25.1)	151 (57.9)	616 (29.1)	
Outside Shanghai	1388 (74.9)	110 (42.1)	1498 (70.9)	
Educational level ^c^ *N* (%)				<0.001
Doctorate	366 (17.7)	12 (4.6)	338 (16.1)	
Master	752 (40.8)	155 (59.4)	907 (43.1)	
Bachelor	765 (41.5)	94 (36.0)	859 (40.8)	
Clinical working experience *N* (%)				0.240
Yes	683 (36.9)	106 (40.6)	789 (37.3)	
No	1170 (63.1)	155 (59.4)	1325 (62.7)	
Physician’s license *N* (%)				0.563
Yes	923 (49.8)	135 (51.7)	1058 (50.0)	
No	930 (50.2)	126 (48.3)	1056 (50.0)	

^a^ WM = Western Medicine; TCM = Traditional Chinese Medicine. ^b^
*p*-values are calculated using the Chi-square test. ^c^ Data were missing for educational level (0.5%).

**Table 3 ijerph-16-03017-t003:** Trainees’ perception of, overall satisfaction toward the SRT program, and change in willingness to become a doctor after the exposure to SRT program (*N* = 2114).

Items	Specialty			*p*-Value ^b^
	Total	WM-Specialty ^a^	TCM-Specialty ^a^	
	*N* = 2114	*N* = 1853 (87.7)	*N* = 261 (12.3)	
Perception of SRT program *N* (%)				<0.001
Very necessary	210 (9.9)	194 (10.5)	16 (6.1)	
Necessary	464 (21.9)	427 (23.0)	37 (14.2)	
Neutral	851 (40.3)	737 (39.8)	114 (43.7)	
Unnecessary	474 (22.4)	397 (21.4)	77 (29.5)	
Very unnecessary	115 (5.4)	98 (5.3)	17 (6.5)	
Overall satisfaction *N* (%)				0.004
Very satisfied	111 (5.3)	104 (5.6)	7 (2.7)	
Satisfied	714 (33.8)	643 (34.7)	71 (27.2)	
Neutral	944 (44.7)	819 (44.2)	125 (47.9)	
Dissatisfied	239 (11.3)	201 (10.8)	38 (14.6)	
Very dissatisfied	106 (5.0)	86 (4.6)	20 (7.7)	
Willingness Change *N* (%)				0.003
Willingness stable/decrease	1729 (81.8)	1534 (82.8)	195 (74.7)	
Willingness increase	385 (18.2)	319 (17.2)	66 (25.3)	

^a^ WM = Western Medicine; TCM = Traditional Chinese Medicine. ^b^
*p*-values are calculated using the Chi-square test.

**Table 4 ijerph-16-03017-t004:** Potential factors associated with trainees’ perception of, overall satisfaction toward the SRT program, and change in willingness to become a doctor after the exposure to SRT program (*N* = 2114) ^d^.

Items	Perception of the SRT Program ^e^		Overall Satisfaction ^e^		Willingness Increase ^e^	
	Univariate Model ^b^	Multivariate Model ^c^	Univariate Model ^b^	Multivariate Model ^c^	Univariate Model ^b^	Multivariate Model ^c^
	Coefficient (95% CI)	Coefficient (95% CI)	Coefficient (95% CI)	Coefficient (95% CI)	OR (95% CI)	OR (95%CI)
Gender						
Female						
Male	−0.06 (−0.15, 0.03)	−0.04 (−0.13, 0.04)	−0.06 (−0.14, 0.02)	−0.06 (−0.14, 0.02)	1.25 (0.99, 1.56)	1.22 (0.96, 1.55)
Age						
	**−0.10 (−0.11, −0.08)**	**−0.03 (−0.06, 0.00)**	**−0.04 (−0.06, −0.02)**	0.00 (−0.03, 0.02)	**1.12 (1.07, 1.17)**	1.06 (0.98, 1.13)
Graduation region						
Outside Shanghai						
Shanghai	**−0.21 (−0.31, −0.12)**	**−0.21 (−0.31, −0.12)**	−0.03 (−0.11, 0.06)	0.00 (−0.08, 0.09)	**1.88 (1.49, 2.36)**	**1.87 (1.46, 2.40)**
Educational level						
Bachelor						
Master	**−0.57 (−0.67, −0.48)**	**−0.39 (−0.55, −0.23)**	**−0.25 (−0.33, −0.17)**	**−0.22 (−0.36, −0.07)**	**1.79 (1.38, 2.31)**	1.08 (0.68, 1.70)
Doctorate	**−0.61 (−0.73, −0.48)**	**−0.33 (−0.55, −0.12)**	**−0.26 (−0.38, −0.15)**	**−0.24 (−0.44, −0.05)**	**2.38 (1.74, 3.27)**	1.16 (0.65, 2.08)
Clinical work experience						
No						
Yes	**−0.23 (−0.32, −0.15)**	−0.03 (−0.13, 0.06)	**−0.10 (−0.18, −0.02)**	−0.02 (−0.10, 0.07)	**1.29 (1.03, 1.62)**	0.96 (0.75, 1.22)
Physician’s license						
No						
Yes	**−0.51 (−0.59, −0.42)**	−0.12 (−0.27, 0.02)	**−0.21 (−0.29, −0.14)**	−0.01 (−0.15, 0.12)	**1.97 (1.57, 2.48)**	**1.64 (1.10, 2.46)**
Specialty						
TCM ^a^						
WM ^a^	**0.28 (0.15, 0.41)**	**0.20 (0.06, 0.33)**	**0.23 (0.11, 0.35)**	**0.23 (0.11, 0.35)**	**0.61 (0.45, 0.83)**	**0.70 (0.51, 0.97)**

^a^ WM = Western Medicine; TCM = Traditional Chinese Medicine. ^b^ Univariate model: Each factor was in the model separately. ^c^ Multivariate model: Factors were all included in the model. ^d^ Bold print indicates statistical significance. ^e^ The perception and overall satisfaction of the SRT program were analyzed by the multiple linear regression model. The willingness increase was analyzed by the multiple logistic regression model.

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
