# Peer review of "The Influence of Standardized Residency Training on Trainees’ Willingness to Become a Doctor: A Comparison between Traditional Chinese Medicine and Western Medicine"

_ijerph, 2019, doi:10.3390/ijerph16173017_

Round 1

Reviewer 1 Report

Thank you for the interesting insights into Chinese residency training. It was an interesting read. To make some issues more clear, I have some suggestions for improvements:

In Table 2 you give p-values, which are related to Chi2 and T-Test significances. This is only clear from the footnote to the table. This is a bit confusing because both methods measure different things. Furthermore, it should be explained in the text what we can learn from these p-values.

In Table 4, you present results from a univariate and a multivariate model. This table is also confusing, because it is not clear what you did there (some points are mentioned in the footnote to the table only). What are "all the other indicators"? What do the numbers in parantheses mean? It would be good to give some more explanations in the text.

Page 8: "In our further study, we found that...": Which further study? 

It is interesting that TCM practioners are less satisfied with the SRT and their willingness to become a doctor is lower, although "The program not only provides more job opportunities for TCM trainees..." (p.8). Do you have explanations why TCM people are less satisfied, although it gives them more job apportunities? 

A proof reading is necessary to correct some minor errors in English.

Author Response

Author's Notes to Reviewer 1

In Table 2 you give p-values, which are related to Chi2 and T-Test significances. This is only clear from the footnote to the table. This is a bit confusing because both methods measure different things. Furthermore, it should be explained in the text what we can learn from these p-values.

We adjusted the table 2 and put the description of age into the text to make the statistical method of table 2 clearer. The detailed methods were also added into the section 2.5 data analyses. You may check the changes on page 4 on line 150-151. Besides, we made some explanations to the findings from p-values. You may check the changes on page 4-5 on line 164-174.

In Table 4, you present results from a univariate and a multivariate model. This table is also confusing, because it is not clear what you did there (some points are mentioned in the footnote to the table only). What are "all the other indicators"? What do the numbers in parantheses mean? It would be good to give some more explanations in the text.

We reinterpreted the footnote to table 4 to make its meaning clearer and also the description in Section 2.5 of the Materials and Methods on page 4 on line 153-157 may help to understand the variables. You may check the changes on page 7 on line 205-209. The numbers in parentheses are 95%CI and we labeled it in the header.

Page 8: "In our further study, we found that...": Which further study?

We have added the relevant reference to illustrate existing research results. You may check the changes on page 10 on line 295.

It is interesting that TCM practioners are less satisfied with the SRT and their willingness to become a doctor is lower, although "The program not only provides more job opportunities for TCM trainees..." (p.8). Do you have explanations why TCM people are less satisfied, although it gives them more job opportunities?

The text here wants to illustrate that the SRT program has explored more possibilities for the development of TCM. The “more job opportunities” mentioned here refers to the fact that TCM general practitioner can work in the community after training which is a new channel for occupation. It does not mean an increase in the number of jobs. To avoid ambiguity, we have made adjustments to the text. You may check the changes on page 10 on line 286.

A proof reading is necessary to correct some minor errors in English.

Thanks for your suggestion. We have checked the grammar of the article and made some modifications.

Reviewer 2 Report

The research question has not been clearly articulated.

The rationale for the study should be made clearer to the reader. 

The description of the SRT program would be better in a figure showing the options as the text is confusing. The figure should be in the methods rather than the introduction.

The duration of the program should also be clarified. Why does someone with a doctorate have a shorter program? What if the doctorate was not relevant to clinical medicine?

The survey tool should be provided in the paper.

The results indicate that only a small percentage of WM and TCM trainees feel the SRT program was necessary or were very satisfied. This makes it challenging to draw any conclusions on the impact of the program suggested by the title. The authors refer to more recent data in the conclusions, but based on the data provided the conclusions are not clear.

There are numerous grammatical and tense issues such as in the following lines:

Line 51: This section I believe should read: "Residency training, regarding as an important part of graduate medical education, was first introduced in China in 1988. Shanghai was... Shanghai has continued..."

Line 107: should be past tense.

Line 137: should not have et al. Not clear what this means.

Line 169: "increased willingness" should state here if this means to practice clinical medicine. The reader should not hav to refer back to figure this out.

This paper would need significant revisions to be considered for publication. The layout needs to be clearer and the rationale for the study more clearly articulated. What is the research question?

Author Response

Author's Notes to Reviewer2

The research question has not been clearly articulated.

The rationale for the study should be made clearer to the reader.

This study aims to explore whether there will be an impact on residents’ willingness to practice medicine in the case of newly-issued SRT policy. The research focuses on the situation at the time of promulgation.

The description of the SRT program would be better in a figure showing the options as the text is confusing. The figure should be in the methods rather than the introduction.

The purpose of this study is to evaluate the impact of the SRT program, so the description of the program is mainly in the background. In the method, we mainly describe the methodology of evaluation.

The duration of the program should also be clarified. Why does someone with a doctorate have a shorter program? What if the doctorate was not relevant to clinical medicine?

Since the residents with master's degree and doctorate already have clinical practice experience during medical school study, the program established a differentiated training time setting for residents with different academic qualifications. We have made some explanation on page 2 on line 62-63 to illustrate this question. Besides, all the residents’ degree is relevant to clinical medicine.

The survey tool should be provided in the paper.

Thanks for your suggestion. The study is aimed to evaluate the trainees’ attitudes toward the SRT program and find out the impact of the SRT program on their willingness to practice medicine. Data on trainees’ attitudes, their satisfaction toward the SRT program, and the change in their willingness to become a doctor after the exposure to the SRT program were collected in June 2014 by an online questionnaire. The questionnaire setting and its distribution were illustrated in section 2.2, 2.3 and 2.4 of Materials and Methods

The results indicate that only a small percentage of WM and TCM trainees feel the SRT program was necessary or were very satisfied. This makes it challenging to draw any conclusions on the impact of the program suggested by the title. The authors refer to more recent data in the conclusions, but based on the data provided the conclusions are not clear.

Thanks for your suggestion. This is a limitation of our study. In this study, we assessed the satisfaction and perception of residents in the initial phase of the program. We hope that we can improve the SRT program based on the results of our research. We also hope that our research can provide a reference for other developing countries.

There are numerous grammatical and tense issues such as in the following lines:

Thanks for your suggestion. We have modified the sentence as the recommendation.

Line 51: This section I believe should read: "Residency training, regarding as an important part of graduate medical education, was first introduced in China in 1988. Shanghai was... Shanghai has continued..."

You may check it on page 2 on line 52-55.

Line 107: should be past tense.

You may check it on page 3 on line 118.

Line 137: should not have et al. Not clear what this means.

Thanks for your question. Since there are 8 TCM sub-specialties and 19 WM sub-specialties, we list part of them and use et al., to represent for the ones have not been listed.

Line 169: "increased willingness" should state here if this means to practice clinical medicine. The reader should not hav to refer back to figure this out.

You may check it on page 5 on line 186.

The layout needs to be clearer and the rationale for the study more clearly articulated. What is the research question?

This study hypothesizes that the implementation of the SRT program may have a negative impact on residents' willingness to practice medicine. This research aims to find out whether the SRT program has an impact on residents’ willingness to practice clinical medicine, especially focusing on the difference between WM and TCM residents. Residents’ perception of, satisfaction toward the SRT program, and the change in their willingness to practice medicine after entering the SRT program were collected to verify the hypothesis. Besides, through the analysis, we hope to find out the factors related to change of the willingness and accordingly make suggestions to promote the SRT program and help reduce the influence. We have modified the text in the introduction to make the research question clearer. You may check it on page 3 on line 92-99.

Round 2

Reviewer 1 Report

Thank you very much for the revised version that answer most of my concerns and comments. There are only minor open points with the data analysis part that should be explained:

Table 2: Instead of T-Test and Chi2, you clearly use now only Chi2, which makes sense for the type of data. But the explanations in the text should be adapted to that (it sounds now as if you have used T-Test). Table 4: In the abstract and section 2.5, you mentioned that you used logistic and linear regression. I still cannot see from the table 4 which type of regression was used where? And why do you need two different type of regressions (plus uni- and multivariate)?

Reviewer 2 Report

The authors have addressed the questions and recommendations. There are still some concerns with the layout of the paper and the conclusions, but the additions clarify much of the previous concerns. I feel the purpose of the paper is much clearer now - there are limitations with this study and further study would be required to clarify the findings. 

Author Response

Thanks for your understanding and supporting.